# Effects of Nordic Walking on Physical Fitness in Patients with Cancer: A Systematic Review

**DOI:** 10.3390/cancers17193170

**Published:** 2025-09-29

**Authors:** Anabel Casanovas-Álvarez, Esther Mur-Gimeno, Jaume Masià Ayala, Carles Fernández-Jané, Raquel Sebio-Garcia

**Affiliations:** 1Grup de Recerca Atenció a la Cronicitat i Innovació en Salut (GRACIS), Universitat Pompeu Fabra, Tecnocampus, Av. d’Ernest Lluch, 32, 08302 Mataró, Spain; acasanovasa@tecnocampus.cat; 2VITAE Escola Universitària de l’Esport, Universitat Abat Oliba (CEU), Balmes 209, 08006 Barcelona, Spain; emurg@uao.es; 3Department of Plastic Surgery, Hospital de la Santa Creu i Sant Pau, 08025 Barcelona, Spain; jmasia@santpau.cat; 4Grup de Recerca en Evidència i Innovació en Fisioteràpia (GREIF), Universitat Pompeu Fabra, Tecnocampus, Av. d’Ernest Lluch, 32, 08302 Mataró, Spain; 5Department of Physical Medicine and Rehabilitation, Hospital Clínic de Barcelona, Carrer Villarroel 170, 08036 Barcelona, Spain; sebio@clinic.cat; 6Fundació Clínic per la Recerca Biomèdica-Institut d’Investigacions Biomèdiques August Pi I Sunyer (FCRB-IDIBAPS), Carrer del Roselló 149-153, 08036 Barcelona, Spain

**Keywords:** Nordic Walking, cancer, exercise, systematic review, meta-analysis, physical fitness, quality of life, physical fitness

## Abstract

**Simple Summary:**

Maintaining regular physical activity during and after cancer treatment can be challenging, despite its known benefits. We investigated whether Nordic Walking, an outdoor exercise using specially designed poles, could offer a practical and engaging way to improve physical fitness in individuals living with and beyond cancer. By systematically reviewing existing clinical trials, we aimed to assess its impact on muscle strength, physical activity levels, quality of life, adherence, and safety. Our findings suggest that Nordic Walking may enhance muscle strength and increase physical activity, particularly among breast cancer survivors, with high adherence rates and no serious adverse effects reported. Although evidence regarding improvements in cardiorespiratory fitness and overall quality of life remains limited, this form of exercise appears to be a feasible and safe option. Further high-quality research is needed to confirm these benefits across a wider range of cancer populations.

**Abstract:**

*Background:* Despite evidence supporting exercise in cancer care, adherence remains low. Nordic Walking (NW), a pole-assisted outdoor activity, may overcome barriers and improve fitness. However, a comprehensive synthesis of its effects on physical fitness in cancer patients is lacking. *Objective:* To evaluate NW’s effects on physical fitness, health-related quality of life (HRQoL), adherence, and safety in patients living with and beyond cancer, compared with no intervention or other exercise programs. *Methods:* This PRISMA-compliant systematic review and meta-analysis (PROSPERO: CRD42024551608) included randomized or quasi-randomized trials. Five databases were searched through November 2024. Risk of bias (Joanna Briggs Institute) and evidence certainty (GRADE) were assessed. *Results:* This systematic review included six RCTs comparing NW with no intervention. NW significantly improved overall muscle strength (Std. MD = 0.46, 95%CI:0.14–0.78; low-certainty) and self-reported physical activity (MD = 3181.51 MET-min/week, 95%CI:2085–4278; moderate-certainty). Cardiorespiratory fitness (6-min walk) showed no significant improvement in random-effects modeling (MD = 84.78 m, 95%CI:−35.6–205.19; very low-certainty). HRQoL data were insufficient for meta-analysis. Adherence exceeded 90% in supervised sessions, with no serious intervention-related adverse events. *Conclusions:* When compared with no intervention NW is feasible and safe, potentially improving muscle strength and physical activity in patients with cancer. Evidence for cardiorespiratory endurance and HRQoL remains inconclusive. To date, no studies have compared NW with other structured exercise programs. Higher-quality RCTs with diverse populations are needed.

## 1. Introduction

Exercise training has become a very important topic of research within the cancer field over the past decades to the point that it is now known to improve overall and disease-free survival. Since the initial randomized controlled trials conducted in the 1990s, several hundred of trials have been published covering virtually almost any cancer type and stage. The evidence supporting the role of exercise in several clinical populations, including patients with cancer is overwhelming [1]. Consequently, leading oncology organizations such as the American Society of Clinical Oncology (ASCO) and others recommend integrating exercise training throughout the cancer care continuum [2].

Despite this endorsement, current data indicate that a large proportion of patients with cancer remain insufficiently active [3]. Numerous barriers have been identified, including a lack of referral or guidance from healthcare providers, low self-efficacy, symptom burden, and logistical challenges such as time constraints or lack of motivation [4,5]. To overcome these barriers, there is an increasing interest in identifying accessible, enjoyable, and effective exercise modalities that can promote adherence while delivering meaningful health benefits.

Nordic Walking (NW) is a modality of training that was initially born as a way of training during summertime for skiers in Nordic countries [6]. This type of exercise encompasses brisk movements of the arms supported by the use of poles to increase the power of walking. The proper technique described by the International Nordic Walking Federation, states that “Nordic Walking is a form of physical activity where natural walking is enhanced by the addition of the active use of pair of specially-designed Nordic Walking poles […] where the use of poles actively engages the upper body into the act of walking to propel the body forward” [7]. Unlike regular walking or running, NW recruits both the upper and lower body, thereby strengthening the arms, shoulders, and trunk in addition to the legs. This broader muscle activation not only increases energy expenditure [8] but also contributes to improvements in upper-body strength and posture [9]. Moreover, NW has been shown to enhance postural stability, stride length, gait pattern, and gait variability in populations with gait instability [10] which may otherwise limit the ability to exercise safely.

Given that NW is a modality of training which involves being outdoors, which is usually preferred by patients, it could help enhance adherence [11]. In addition, most patients with cancer refer walking as their preferable way of exercise [12], thus NW could be seen as a way of increasing both intensity and adherence to exercise in this population. Furthermore, its potential to combine safety, accessibility, and comprehensive fitness benefits makes NW an attractive option within oncology rehabilitation and survivorship care.

However, despite its potential, there is a lack of comprehensive synthesis specifically evaluating the effects of NW on fitness-related outcomes such as cardiorespiratory endurance, muscular strength, and balance in cancer patients. These fitness components are critical for treatment tolerance, functional independence, and overall quality of life. Moreover, only one prior systematic review has synthesized findings related to NW in breast cancer which is now outdated, having been published over five years ago [13]. Given the likely emergence of new evidence across diverse cancer types and fitness-related outcomes, an updated and broader review is warranted.

Therefore, the aim of this systematic review and meta-analysis was to synthesize the current body of evidence on the effects of NW in individuals with cancer, with a specific focus on outcomes related to physical fitness (including cardiorespiratory endurance, muscle strength, and balance), health-related quality of life (HRQoL), adherence, and safety.

## 2. Materials and Methods

This systematic review adheres to the Preferred Reporting Items for Systematic Reviews and Meta-Analyses (PRISMA) guidelines to ensure transparency, reproducibility. The protocol was pre-registered with the International Prospective Register of Systematic Reviews (PROSPERO; registration number CRD42024551608).

### 2.1. Inclusion and Exclusion Criteria

The eligibility criteria for studies were established using the PICO framework (Population, Intervention, Comparator, Outcome), as detailed below:Design: Only randomized controlled trials (RCTs) or quasi-randomized controlled trials (quasi-RCTs) were included to ensure high-quality evidence. Non-controlled studies, observational studies, case reports, and qualitative research were excluded.Population: Studies involving participants diagnosed with any type of cancer at any stage were included. There were no restrictions regarding age, sex, or ethnicity. Studies focusing on non-oncological populations or mixed populations where cancer-specific outcomes could not be extracted were excluded.Intervention: The intervention of interest was NW either as a standalone intervention or in combination with usual care. Eligible studies had to provide a structured NW program alone or in combination with other exercises.Comparator: At least one of the following comparators was required: Nordic Walking versus no intervention (this included usual care and general exercise recommendations but excluded specific exercise programs). Nordic Walking versus another structured exercise program.

Studies without a comparator or with comparators irrelevant to the research question were excluded.
Outcomes: Studies were required to report at least one of the following outcomes: Cardiorespiratory endurance: Assessed via field tests (e.g., six-minute walk distance [6MWD]) or laboratory-based measures (e.g., VO_2_ max). Muscle strength: Evaluated using direct measures (e.g., handgrip dynamometry) or functional assessments (e.g., sit-to-stand [STS] test). Balance: Quantified through balance assessments (e.g., Berg Balance Scale or timed up-and-go test) or equivalent tools. Physical Activity level: Assessed through self-reported questionnaires (e.g., International Physical Activity Questionnaire [IPAQ]) or objective measures (e.g., accelerometry). Health Related Quality of Life (HRQoL): Evaluated using validated questionnaires specific to oncological or general populations (e.g., 36 Health Survey [SF-36]). Adherence: Evaluated by the number or proportion of participants who completed the intervention period. Safety: Evaluated by the number and type of adverse events.

Exclusion criteria: Secondary research, including systematic reviews and study protocols, were excluded from the analysis. However, systematic reviews were screened for relevant primary studies that met the inclusion criteria. There were no initial restrictions on language or year of publication, allowing for a comprehensive and inclusive review of the literature.

### 2.2. Information Sources and Search Strategies

The systematic review was conducted using comprehensive searches in PubMed, MEDLINE (via Ovid), Cochrane Library Plus, Web of Science, and PEDro (Physiotherapy Evidence Database). Access to these databases was provided through the University of Barcelona (UB) and TecnoCampus. The initial search was performed in October 2023, with an updated search conducted in November 2024 to include the most recent studies.

Full details of the specific search equations used in each database are available in the Appendix A, ensuring transparency and reproducibility. To further ensure the comprehensiveness of the search, systematic reviews on related topics were reviewed to identify potentially eligible primary studies.

### 2.3. Study Selection

The study selection process was managed using the Rayyan^®^ software platform [14], which facilitated the detection and removal of duplicate records and streamlined collaboration among reviewers. Study selection was conducted in two stages. In the first stage, titles and abstracts were screened for eligibility based on the inclusion criteria. In the second stage, full texts of potentially eligible studies were assessed for inclusion.

Two independent reviewers (AC, EM, AC, and CF) performed the screening, and disagreements were resolved by discussion. If consensus could not be reached, a third reviewer acted as an arbitrator. All reasons for excluding studies at the full-text stage were recorded in the Rayyan^®^ platform. A complete list of excluded studies and their reasons for exclusion are provided in the Appendix A.

### 2.4. Data Extraction

Relevant data from each included study were extracted and recorded independently by pairs of reviewers using Microsoft Excel^®^ version 16 (Microsoft^®^, IBM, USA) to minimize errors. The extracted information included the following: Publication Characteristics (Author, year, and country of publication); Study Design (Number of arms, parallel or crossover, and whether the trial was single-center or multicenter); Participant Characteristics (Number of participants, sex distribution, age, cancer type, phase of therapy, e.g., off treatment, maintenance, acute treatment, or mixed, and other relevant characteristics); Intervention (Setting, adjuvant treatments, frequency, duration, number of sessions, intensity, progression, and supervision); Comparator (Type of comparator, e.g., no intervention, usual care, or exercise and, in the case of exercise, a description of the intervention including setting, adjuvant treatments, frequency, duration, number of sessions, intensity, progression, and supervision); Outcomes (List of outcomes assessed, measurement instruments used, and time points of assessment); Results (Descriptive data for each study group, outcome, and time point, e.g., mean and standard deviation as well as comparative data between groups, e.g., mean difference, confidence intervals, and *p*-values).

A pilot test of the data extraction form was conducted before formal data extraction to ensure its usability and comprehensiveness. Any discrepancies between reviewers were resolved by consensus through discussion.

### 2.5. Risk of Bias for Individual Studies

The methodological quality of the included studies was evaluated using the Joanna Briggs Institute (JBI) tool for randomized controlled trials [15]. This tool includes 13 questions addressing critical aspects of study design and methodology. Each question was assessed using the following options: Yes, No, Unclear, or Not Applicable. The domains evaluated encompassed the randomization process, allocation concealment, baseline similarity of groups, blinding of participants and those delivering the intervention, blinding of outcome assessors, and equality of treatment between groups beyond the intervention. Other aspects assessed included completeness and differences in follow-up, application of intention-to-treat analysis, consistency and reliability of outcome measurement, appropriateness of statistical analysis, and overall suitability of the study design.

Two independent reviewers (EM and AC) performed the assessments. In cases of discrepancies, reviewers engaged in discussions to achieve consensus. If consensus could not be reached, a third reviewer was consulted to resolve disagreements.

### 2.6. Measurements of Treatment

For all outcomes, we used data at the end of each specific timepoint of each group, only if this was not available, did we use the group change from baseline.

The unity of analysis was individual participants. For efficacy outcomes, we prioritized data from the intention to treat analysis. In case of missing data, we tried to contact the authors, if no answer was given, that particular study was not included in the analysis.

#### 2.6.1. Efficacy Outcomes

The primary efficacy outcome was cardiorespiratory fitness assessed at the end of the intervention. Secondary outcomes included cardiorespiratory fitness at three- and six-months post-intervention, muscle strength, balance, physical activity level, HRQoL, adherence, and safety assessed at the end of the intervention, three months, and six months post-intervention.

For all outcomes, when a composite score consisting of multiple subindexes was reported, the total score was used whenever available. For strength outcomes, a global score was prioritized. If only data on specific isolated movements were available, a global estimate was calculated by averaging the values across these movements. This approach ensures that studies reporting multiple strength measures do not disproportionately influence the meta-analysis, improves comparability across studies that assess different strength tests, and captures the overall functional impact of muscle strength. Additionally, it simplifies statistical analysis by reducing the number of correlated outcomes and maintains interpretability, providing a single, clinically meaningful measure for readers and clinicians. For HRQoL of life, when the SF-36 was used, the Physical Component Summary (PCS) was included in the metanalysis where possible. If PCS was not directly reported, it was derived from the individual domain scores using established weighting algorithms. In cases where only domain-level data were available and conversion to a summary score was not feasible, these results were summarized qualitatively.

#### 2.6.2. Adherence and Safety Outcomes

Adherence was measured as the percentage of planned intervention sessions attended by participants. Safety was evaluated based on the number of participants who experienced at least one adverse event during the study period.

### 2.7. Data Synthesis

Data were analyzed for each comparison outlined in the review (NW vs. no intervention and NW vs. other exercise programs) and for three time points: post-intervention, three months post-intervention, and six months post-intervention. Studies not matching these exact time points were included in the closest category.

When sufficient data were available and studies were deemed homogeneous in terms of participants, interventions, and control groups, results were synthesized using meta-analysis. In cases where homogeneity was not achieved, a narrative synthesis was performed instead.

Meta-analyses were conducted using Review Manager (RevMan^®^) software, version 5.4.1 [16]. Continuous outcomes were analyzed using mean differences (MD) when identical measurement tools were employed across studies, and standardized mean differences (Std. MD) when tools differed. Dichotomous outcomes were assessed using risk ratios (RR) and absolute risk differences (ARD). Due to expected variability in the interventions, a random-effects model was used to calculate pooled effect sizes with 95% confidence intervals (95%CI). Results were presented visually using forest plots to enhance clarity. Heterogeneity was assessed using the I^2^ statistic. Thresholds for interpretation were as follows: I^2^ < 30%: Low heterogeneity; I^2^ between 30% and 70%: Moderate heterogeneity; I^2^ > 70%: Substantial heterogeneity.

Subgroup analyses were planned to explore the effects of cancer type, strength (hand grip, upper extremity, trunk, and lower extremity), treatment duration, and total number of sessions. Sensitivity analyses were conducted systematically by excluding one study at a time to evaluate whether any single study disproportionately influenced the overall results.

### 2.8. Publication Bias

When sufficient data were available, defined as a minimum of ten studies per analysis, publication bias was assessed using funnel plots. Visual inspection of asymmetries in the funnel plots was employed to identify potential publication bias.

### 2.9. Summary of Finding and Assessment of the Certainty of the Evidence

To assess the quality of evidence for each outcome, we used the Grading of Recommendations Assessment, Development and Evaluation (GRADE) framework [17]. The GRADE approach evaluates the certainty of evidence based on the following domains: Risk of Bias: Assessed using the JBI tool; Inconsistency: Evaluated through heterogeneity across study results; Indirectness: Examined by determining whether the evidence directly applies to the population, intervention, and outcomes of interest; Imprecision: Assessed based on confidence intervals and overall sample size; Publication Bias: Considered through funnel plot assessments.

Each outcome was rated as having high, moderate, low, or very low certainty based on these domains. The GRADE approach allowed for a comprehensive summary of the evidence, highlighting the strength of the findings and areas requiring further investigation.

## 3. Results

### 3.1. Search Results

The final database search was conducted on 8 November 2024, yielding a total of 60 references. After removal of duplicates, 32 unique articles remained for screening based on titles and abstracts. Of these, 10 full-text articles were retrieved for detailed eligibility assessment. Following application of the predefined inclusion and exclusion criteria, six articles met eligibility criteria and were included in the systematic review [18,19,20,21,22,23].

A detailed flowchart of the study selection process is presented in Figure 1. A comprehensive list of excluded studies, along with reasons for exclusion, is provided in the Appendix A.

### 3.2. Characteristics of the Studies Included

The characteristics of the included studies are summarized in Table 1.

#### 3.2.1. Publication Characteristics

The six included randomized controlled trials (RCTs) were published between 2011 and 2024. The majority were conducted in European countries, including Spain [18], the United Kingdom [20], Poland [21,22], and Germany [23], with one study conducted in Canada [19]. All trials employed a two-arm, parallel-group, single-center design. Two of the studies were identified as pilot trials [19,20].

#### 3.2.2. Participant Characteristics

Sample sizes ranged from 8 to 61 participants. One pilot study included only 8 participants [19]. The mean age of participants across studies ranged from 49.2 to 67 years. All studies included female participants, with one study [19] also including two male participants. Five studies [18,20,21,22,23] enrolled exclusively breast cancer patients, and one study [19] included participants with various cancer types (lung, prostate, colorectal, and endometrial cancers). Regarding phase of therapy, two studies [18,23] included participants undergoing active treatment; one study [20] focused on patients in the maintenance phase receiving aromatase inhibitors; two studies [21,22] enrolled participants in the post-treatment phase (≥1 year post-surgery); and one study [19] included a mixed group of participants either in treatment, post-treatment, or with no prior treatment.

#### 3.2.3. Intervention Characteristics

Detailed intervention protocols are summarized in Table 2. Four studies [19,20,22,23] investigated NW as a standalone intervention, while two studies combined NW with additional modalities: bodyweight resistance exercises [18] or general gymnastics [21]. Intervention durations ranged from 4 to 16 weeks, encompassing 16 to 34 sessions in total. Most studies prescribed two sessions per week, with frequencies ranging from one to four sessions weekly.

Supervision formats varied: four studies delivered fully supervised group sessions [18,21,22,23], while two studies incorporated both supervised group sessions and unsupervised home-based sessions [19,20]. Session durations typically ranged from 45 to 60 min; one study employed 75-min sessions [18], while another reported variable session lengths (20 to 60 min) based on participants’ baseline physical activity levels [19]. All interventions included standardized warm-up and cool-down periods. Among studies reporting exercise intensity, moderate to vigorous intensity was most frequently prescribed.

#### 3.2.4. Control Group Characteristics

In all studies, the addition of NW to usual care was evaluated. Control conditions consisted of usual care alone [18,19,22,23] or general gymnastics alone [21], depending on study design. One study [20], employed an enhanced usual care comparator, which included biweekly counseling sessions and provision of a physical activity informational booklet. No included study directly compared NW to other structured exercise interventions.

**Table 1 cancers-17-03170-t001:** Characteristics of the included studies.

Autor Year Country	Study Design	Participants Sample Size Male/Female Age (Mean and SD) Cancer Type Phase of Therapy Others	Intervention	Control	Outcomes and Measurement Tools	Endpoints
Casanovas-Álvarez 2024 Spain [18]	2-arm parallel RCT Single-center	61 (30 IG, 31 CG) All female IG: 49.2 (10.9), CG: 54.7 (12.1) Breast cancer In treatment (Scheduled for tumor resection)	Nordic Waling and body weight exercises	Usual care	Cardiorespiratory endurance – Lower body (6MWT) Muscle strength: – Handgrip strength (dynamometer) Physical activity level (IPAQ) HRQoL (EORTC-QLQC30) Adherence (% of session attendance) Safety (number of events)	Post intervention 1-month post intervention 3-months post intervention
Cunningham 2020 Canada [19]	2-arm pilot parallel RCT Single-center	8 (4 IG and 4 CG) 2 male and 6 female 67 (5.8) Mixed cohort (Lung, Prostate, Colorectal, Endometrial) In treatment or post-treatment: 7; No treatment: 1	Nordic walking	Usual care	Cardiorespiratory endurance (6MWT) Muscle strength – Handgrip strength (NR) – Lower body strength (30-s CTS) Physica activity level (IPAQ) HRQoL (SF-36) Adherence (number of participants completing the program) Safety (NR)	Post intervention
Fields 2016 UK [20]	2-arm pilot parallel RCT Single-center	40 (20 IG and 20 CG) All female 63 (8) Breast Cancer Maintenance phase With aromatase inhibitor-associated arthralgia	Nordic Walking only	Enhanced usual care (contacted every 2 weeks and booklet on physical activity)	Physical activity level (GPPAQ) HRQoL (SF-36) Adherence (% of session attendance) Safety (Injury prevalence and type/outcome)	Post intervention
Hanuszkiewicz 2021 Poland [21]	2-arm parallel RCT Single-center	39 (19 IG and 20 CG) All female 58.8 (7.3) Breast Cancer Post-treatment (≥1 year post-surgery)	Nordic walking and general gymnastics training	General gymnastics based on guidelines for cancer survivors	Muscle strength: – Endurance of the trunk muscles (Biodex Multi-Joint 3 Isokinetic Dynamometer)	Post intervention
Malicka 2011 Poland [22]	2-arm parallel RCT Single-center	38 (23 IG and 15 CG) All female 62.8 (6.1) Breast Cancer Pot-treatment (average time since surgery 7.6 years)	Nordic Walking only	Usual care	Muscle strength: – Upper extremity strength (Biodex Multi Joint 3 isokinetic dynamometer)	Post intervention
Rösner 2011 Germany [23]	2-arm parallel RCT Single-center	50 (26 IG and 24 CG) IG 52.4 (5.5), CG: 50.8 (5.9) All female Breast Cancer In treatment (Initial diagnosis of tumor stages T1–3, N0-1, and M0)	Nordic Walking only	Usual care	Muscle strength: – Shoulder and elbow isometric maximum strength (dynamometer) HRQoL (SF-12)	Post intervention

30-s CTS: 30-s chair stand test; 6MWT: Six-Minute Walk Test; EORTC-QLQC30: European Organization for Research and Treatment of Cancer Quality of Life; GPPAQ: GP Physical Activity Questionnaire; HRQoL: Health Related Quality of Life; IPAQ: International Physical Activity Questionnaire; SF-12: 12-item Short-Form Health Survey; SF-36: 36-item Short-Form Health Survey.

**Table 2 cancers-17-03170-t002:** Description of Nordik Walking interventions.

Study	Nº Sessions and Supervision	Frequency	Session Description	Intensity	Progression	Session Duration	Length of Intervention
Casanovas-Álvarez 2024 [18]	From 24 to 32 supervised group sessions	Twice per week	Warm up (10 min) NW (10 min) Body weight exercises (10 exercises, 15–30 rep per exercise) NW (15 min) Body weight exercises Cool-down (15 min)	Moderate-to-vigorous (RPE of 6–8)	Body weight exercise repetitions were increased every 2 weeks	75 min	12 to 16 weeks
Cunningham 2020 [19]	8 supervised group session (1 session/week) 24 non-supervised individual session (up to 3 sessions/week)	One up to four times per week	Individualized prescription depending on the PA level. High active: NW (30–60 min) Minimally active: NW (30–45 min) Inactive: NW (20–30 min)	Moderate intensity (Borg 11–15)	Time was increased from 20–30 min to 60	20–60 min	8 weeks
Fields 2016 [20]	6 Supervised group session (first 6 weeks) Up to 12 non-supervised individual session	One a week for the first 3 weeks and increasing to 2, 3 and 4 times a week every two weeks	Warm-up (10 min) NW (30 min) + Cool-down (10 min)	Moderate intensity (Borg 11–13)	Intensity was increased progressively using on Borg scale	50 min	12 weeks
Hanuszkiewicz 2020 [21]	16 supervised group sessions	Twice per week	Warm-up (5 min) NW (35 min) Cool-down (5 min)	65–70% of maximal HR	Intensity was increased gradually by decreasing the number of rest periods.	45 min	8 weeks
Malicka 2011 [22]	16 supervised group sessions	Twice per week	Warm up (10 min) NW (40 min) Cool-down (10 min)	Up to 85% of maximal HR	NR	60 min	8 weeks
Rösner 2011 [23]	12 sessions, supervision NR	Tree times per week	Warm-up (NR) Playful forms (NR) Walking without poles (NR) Stretching exercises leg muscles and upper body (NR)	NR	NR	60 min	4 weeks

HR: Heart Rate; NW: Nordic Walking; NR: Not Reported; RPE: Rating of Perceived Exertion.

### 3.3. Risk of Bias Assessment

The details for the risk of bias assessment can be found in Table 3.

Overall, the methodological quality of the included studies was rated as low to moderate. Due to the nature of the intervention, blinding of participants and intervention providers was not feasible in any of the trials. Additionally, outcome assessors were not blinded in three [19,20,22] of the six studies, and in the remaining three [18,21,23], the blinding status was unclear due to insufficient reporting.

Incomplete reporting also affected other key domains. In particular, random sequence generation was inadequately described in three studies [21,22,23], and allocation concealment was unclear in four studies [20,21,22,23], limiting confidence in the internal validity of these trials.

**Table 3 cancers-17-03170-t003:** Risk of bias assessment.

	Casanovas-Álvarez 2024 [18]	Cunningham 2020 [19]	Fields 2016 [20]	Hanuszkiewicz 2020 [21]	Malicka 2011 [22]	Rösner 2021 [23]
Item 1	YES	YES	YES	UNCLEAR	UNCLEAR	UNCLEAR
Item 2	YES	YES	UNCLEAR	UNCLEAR	UNCLEAR	UNCLEAR
Item 3	YES	UNCLEAR	YES	YES	YES	YES
Item 4	NO	NO	NO	NO	NO	NO
Item 5	NO	NO	NO	NO	NO	NO
Item 6	UNCLEAR	NO	NO	UNCLEAR	NO	UNCLEAR
Item 7	YES	UNCLEAR	YES	YES	YES	YES
Item 8	YES	YES	YES	YES	UNCLEAR	YES
Item 9	YES	YES	UNCLEAR	YES	UNCLEAR	YES
Item 10	YES	YES	YES	YES	YES	YES
Item 11	YES	YES	YES	YES	YES	YES
Item 12	YES	YES	YES	YES	YES	YES
Item 13	YES	YES	YES	YES	UNCLEAR	YES

**Item 1:** Was true randomization used for assignment of participants to treatment groups? **Item 2:** Was allocation to treatment groups concealed? **Item 3:** Were treatment groups similar at the baseline? **Item 4:** Were participants blind to treatment assignment? **Item 5:** Were those delivering treatment blind to treatment assignment? **Item 6:** Were outcomes assessors blind to treatment assignment? **Item 7**: Were treatment groups treated identically other than the intervention of interest? **Item 8:** Was follow up complete and if not, were differences between groups in terms of their follow up adequately described and analyzed? **Item 9:** Were participants analyzed in the groups to which they were randomized? **Item 10:** Were outcomes measured in the same way for treatment groups? **Item 11:** Were outcomes measured in a reliable way? **Item 12**: Was appropriate statistical analysis used? **Item 13**: Was the trial design appropriate, and any deviations from the standard RCT design (individual randomization, parallel groups) accounted for in the conduct and analysis of the trial? Green = Yes, yellow = Unclear, and red = No.

### 3.4. Efficacy Outcomes

A summary of all efficacy outcomes assessed in each study, along with the corresponding measurement instruments and assessment time points, is presented in Table 1.

#### 3.4.1. Nordic Walking vs. No Intervention

##### Cardiorespiratory Endurance

Two studies, encompassing a total of 68 participants, evaluated cardiorespiratory fitness using the 6-Minute Walking Test (6MWT) [18,19].

For post-intervention effects, the meta-analysis using a random-effects model demonstrated no statistically significant difference between the NW and control groups (MD: 84.78 m; 95%CI: −35.6 to 205.19; I^2^ = 51%) (Figure 2). However, when analyzed using a fixed-effects model, a statistically significant improvement favoring the intervention group was observed (MD: 114.35 m; 95%CI: 66.42 to 162.27; I^2^ = 51%) (Appendix A). In both analyses, moderate heterogeneity was present.

At 3 months post-intervention, only the study by Casanovas-Alvarez et al. [18], assessed cardiorespiratory fitness. The authors reported a statistically significant improvement favoring the NW group (613.53 m vs. 526.4 m; t = 4.097; *p* < 0.001). Notably, the intervention group also demonstrated superior baseline values (592.13 m vs. 537.9 m; t = 2.556; *p* = 0.001).

No studies assessed cardiorespiratory fitness at 6 months follow-up.

##### Muscle Strength

Muscle strength was assessed in five studies post-intervention [18,19,21,22,23], including handgrip, upper body, trunk, and lower body strength. Only four studies provided sufficient data for inclusion in the meta-analysis, as Rösner et al. [23] did not report extractable numerical values.

The meta-analysis, which included 156 participants, demonstrated a statistically significant improvement in overall muscle strength favoring NW (Std. MD: 0.46; 95%CI: 0.14 to 0.78; I^2^ = 0%) (Figure 3). Subgroup analysis for handgrip strength, which included data from multiple studies, revealed no statistically significant difference (Std. MD: 0.39; 95%CI: −0.08 to 0.86; I^2^ = 0%) (Figure 3). The results remained consistent when analyzed using a fixed-effects model (Appendix A).

Sensitivity analysis indicated that the overall effect became non-significant only when excluding the study by Hanuszkiewicz et al. [21] (Appendix A).

Rösner et al. [23] reported significant increases in shoulder strength favoring the intervention group (*p* < 0.05); however, only graphical data were presented, and no numerical data could be extracted. Attempts to contact the authors for additional data were unsuccessful.

At 3 months post-intervention, only Casanovas-Alvarez et al. [18] assessed handgrip strength, reporting no statistically significant difference between groups (19.6 kg vs. 19.8 kg; *p* ≥ 0.05).

##### Balance

No studies assessed balance outcomes at any time point.

##### Physical Activity Level

Physical activity was assessed in three studies post-intervention [18,19,20]. However, the study by Fields et al. [20] was not included in the meta-analysis, as it used a categorical approach based on the GP Physical Activity Questionnaire, which was not comparable to the continuous data reported in the other studies.

The meta-analysis of the remaining two studies, comprising 67 participants, demonstrated a statistically significant improvement in physical activity levels as measured by the International Physical Activity Questionnaire (IPAQ) (MD = 3173.05 MET-min/week; 95% CI: 2076.61 to 4269.49; I^2^ = 0%) (Figure 4a). Due to the wide range of values observed, an additional analysis using Std.MD was performed for improved visualization and interpretability (Figure 4b), which yielded consistent results. Findings remained unchanged when analyzed using a fixed-effects model (Appendix A).

Fields et al. [20] reported categorical changes in physical activity: 39% (7/18) of participants in the NW group reported increased vigorous physical activity, with no change observed in walking activity. In the control group, 45% (9/20) reported an increase in walking activity, and 15% (3/20) reported an increase in vigorous activity.

##### HRQoL

HRQoL was assessed in four studies post-intervention [18,19,20,23]. However, meta-analysis could not be performed due to variability in reporting and lack of sufficient quantitative data. Specifically, Cunningham et al. [19], Fields et al. [20], and Rösner et al. [23] utilized the SF-36 questionnaire but did not report the Physical Component Summary (PCS) scores nor sufficient domain-level data to allow for calculation of a composite score.

Casanovas-Alvarez et al. [18] employed a cancer-specific HRQoL questionnaire (the European organization for research and treatment of cancer quality life [EORTC-QLQC30]) and reported statistically significant between-group differences favoring the NW group (MD: 6.7 points; 95% CI: 0.1 to 13.4).

Among the studies using the SF-36, Cunningham et al. [19] reported no statistically significant differences between groups across any domains. Fields et al. [20] described a general trend toward improvement in the intervention group; however, no between-group statistical comparisons were reported. Rösner et al. [23] presented only *p*-values, reporting significant within-group improvements in most SF-36 domains for the NW group, except for “Mental Health” and “General Health Perception,” but no direct between-group comparisons were provided.

At 3-month follow-up, only Casanovas-Alvarez et al. [18] assessed HRQoL and found no statistically significant differences between groups.

#### 3.4.2. Nordic Walking vs. Other Exercise Programs

No studies directly compared Nordic Walking to other structured exercise interventions.

### 3.5. Adherence and Safety

Three studies [18,19,20] reported adherence outcomes and adverse events. Due to heterogeneity in how adherence was measured and safety data were reported, a meta-analysis was not possible, and results are summarized narratively.

#### 3.5.1. Adherence

Adherence to the NW intervention was generally high across studies.

Casanovas-Alvarez et al. [18] reported a mean adherence of 92.2%, with participants attending a median of 16 sessions over 11 weeks.

Cunningham et al. [19] reported that 3 out of 4 participants completed the full NW program; one participant withdrew due to personal reasons.

Fields et al. [20] reported 90% adherence to supervised sessions, with a median of 5 out of 6 sessions attended per participant. Adherence to the unsupervised component was lower; most participants completed 1 to 2 sessions per week, and only 8% met the prescribed four sessions weekly.

#### 3.5.2. Safety

Three studies reported on safety outcomes. Casanovas-Alvarez et al. [18] and Cunningham et al. [19] reported no adverse events related to the NW intervention. Fields et al. [20] reported that 6 out of 20 participants (30%) experienced pain during the study; in four cases, symptoms were pre-existing. Most issues resolved with physiotherapy, except one case involving a new cancer diagnosis. No participants developed new lymphedema, and all those with pre-existing lymphedema (15%; 3/20) showed improvement after the intervention.

### 3.6. Reporting Bias

Due to the limited number of trials included in each meta-analysis (fewer than 10 studies), formal assessment of publication bias using funnel plots or other methods was not feasible.

### 3.7. Cartain of the Evidence

A summary of the certainty of the evidence is provided in the Appendix A. Certainty was assessed using the GRADE framework for the three outcomes for which meta-analyses could be conducted: cardiorespiratory endurance, muscle strength, and physical activity level.

Cardiorespiratory endurance was rated as very low certainty, due to concerns about risk of bias (lack of assessor blinding), inconsistency (differences between fixed- and random-effects models), and imprecision (wide confidence intervals).

Muscle strength was rated as low certainty, downgraded for risk of bias and imprecision, primarily due to potential detection bias and small sample sizes.

Physical activity level was rated as moderate certainty, with downgrading for risk of bias due to reliance on self-reported data via the IPAQ without participant blinding.

## 4. Discussion

This review is the first since 2019 to focus specifically on the fitness-related effects of NW in patients living with and beyond cancer. Across six RCTs, NW produced moderate improvements in global muscle strength and increased self-reported physical activity levels, while cardiorespiratory fitness showed inconsistent results, depending on the statistical model used.

Recent meta-analyses confirm that both resistance training and aerobic exercise independently improve muscle strength, cardiorespiratory fitness, and HRQoL in cancer patients [24]. However, conventional exercise programs frequently encounter adherence barriers, including limited accessibility and low motivation. NW may help address these challenges. As a whole-body activity that combines aerobic movement with upper-body engagement through pole use, NW has been shown to yield similar or even superior improvements in strength and functional capacity compared to standard walking or traditional strength training [8,25]. Given that walking is the most commonly preferred activity among cancer survivors [12], the additional use of poles enhances both exercise intensity and enjoyment without increasing perceived exertion. These characteristics make NW a patient-centered, pragmatic option that could overcome key participation barriers while delivering clinically meaningful fitness benefits.

Although the meta-analysis for cardiorespiratory fitness did not reach statistical significance, the observed mean difference was 77.57 m, which exceeds the minimum clinically important differences (MCIDs) previously reported for patients with cancer. Specifically, MCIDs for the 6MWD have been estimated at 66.5 and 41.5 m in patients undergoing chemotherapy, and 41.4 and 40.5 m in patients after treatment [26]. These findings suggest that NW may lead to clinically meaningful improvements in functional capacity, even if statistical significance was not achieved due to limited sample sizes and wide confidence intervals.

Similarly, the pooled effect size for muscle (Std. MD: 0.46) represents a moderate effect, reinforcing NW’s potential to support muscular conditioning in people with cancer. These findings are consistent with evidence from older adults and individuals with cardiac or metabolic conditions [27,28], where the combination of upper-body engagement and increased energy expenditure has been shown to enhance both strength and activity behavior.

Regarding the physical activity level, the observed increase of over 3000 MET-min/week in self-reported physical activity represents a large and clinically meaningful change, corresponding to a shift into the ‘high activity’ category according to IPAQ classification [29]. This suggests that Nordic Walking may effectively promote significant increases in physical activity levels among cancer survivors.

However, evidence for HRQoL benefits remains tentative, and no data were available regarding balance outcomes. While some studies suggested trends toward improved HRQoL following NW interventions, these findings did not consistently reach statistical significance. Importantly, trials that reported no significant between-group differences in HRQoL were pilot studies [19,20] with limited sample sizes and insufficient statistical power to detect meaningful changes. This highlights a need for larger, adequately powered trials to more definitively assess HRQoL effects. Given that improvements in physical function, muscular strength, and physical activity levels—domains closely linked to quality of life [30,31]—were observed, it is plausible that NW may have a positive impact on HRQoL over time. However, the current evidence base remains too limited to draw firm conclusions.

Adherence to supervised sessions exceeded 90% in two of the three studies—a figure comparable to, or better than, other exercise programs in cancer care [32,33]. Importantly, no serious adverse events attributable to NW were reported.

These findings broadly corroborate the conclusions of the earlier systematic review by Sánchez-Lastra et al. [13], which focused solely on breast cancer survivors and included both RCTs and quasi-experimental studies. That review also highlighted potential improvements in muscle strength and physical activity, as well as high adherence and safety profiles, particularly in supervised settings. However, unlike our review, it reported more consistent positive effects on upper limb strength and lymphedema-related outcomes—likely due to its inclusion of non-randomized designs and studies with a narrower population focus. Moreover, while Sánchez-Lastra et al. identified improvements in disability and morbidity perceptions, our review did not include such endpoints, and found no studies assessing balance. Conversely, our review adds updated evidence on HRQoL and the medium-term effects of NW on physical activity behavior—domains that were either not explored or reported narratively in the previous review.

### 4.1. Limitations

Several constraints should be acknowledged. First, the number of studies specifically targeting physical fitness outcomes in cancer patients undergoing NW interventions remains limited, restricting the depth and generalizability of the findings. Only three meta-analyses could be performed, and two of them—cardiorespiratory endurance and physical activity level—each included just two studies, one of which was a pilot trial with only eight participants. The muscle strength meta-analysis included four studies, but most subgroup analyses (e.g., upper body, trunk, lower body strength) were based on single studies, and only handgrip strength was assessed in more than one study.

This small and heterogeneous evidence base not only reduces the precision of the effect estimates but also limits the ability to conduct robust sensitivity analyses or assess publication bias. Most of the included studies were small, single-center trials and demonstrated a moderate overall risk of bias, with key concerns identified in several methodological domains. In particular, allocation concealment was unclear in the majority of studies, and blinding of participants and intervention providers was not feasible, a limitation inherent to the nature of exercise interventions such as Nordic Walking. However, blinding of outcome assessors is feasible and should be implemented in future trials to reduce detection bias.

Moreover, outcome reporting was often incomplete or insufficiently detailed, especially regarding randomization procedures and blinding. These limitations highlight the need for greater adherence to established reporting standards, such as the CONSORT guidelines [34], to ensure methodological transparency and reproducibility.

Finally, the predominance of studies conducted in breast cancer populations, along with the absence of trials involving other cancer types or treatment stages, prevented exploration of differential effects across cancer subgroups. As a result, the applicability of the findings to the broader oncology population remains limited.

### 4.2. Implications for Practice, Policy and Future Research

Current evidence supports NW as a safe, acceptable adjunct to usual care or survivorship programs, particularly for women treated for breast cancer who may benefit from upper extremity loading [13]. Clinicians can consider recommending supervised NW when resources allow, with realistic expectations that the largest short-term gains are likely in muscle strength parameters and overall activity volume rather than in cardiorespiratory capacity.

Exercise oncology guidelines [2,35] already advocate moderate intensity aerobic activity; recognizing NW as a structured option could help diversify offerings where walking is the preferred modality and pole training expertise is available. Integrating NW into community cancer rehabilitation networks may improve reach and adherence.

Future trials should aim to recruit more heterogeneous cancer cohorts, including male participants, to improve generalizability. There is also a need to standardize core outcome sets to facilitate pooling of results across studies. Comparative trials evaluating NW against other aerobic or multimodal interventions, ideally using non-inferiority designs, would help delineate its unique benefits. In addition, extending follow-up beyond six months and monitoring the maintenance of unsupervised activity is essential, given the observed decline in adherence once supervision ended. Finally, future research should employ robust methodological safeguards, including proper allocation concealment, assessor blinding, and pre-registered protocols, and report findings in accordance with CONSORT guidelines.

## 5. Conclusions

Nordic Walking appears to be a feasible and low-risk exercise option that may improve muscle strength (low-certainty evidence) and physical activity levels (moderate-certainty evidence) in people with cancer when compared with no intervention. However, current evidence is too limited, heterogeneous, and largely concentrated in breast cancer populations to draw firm conclusions regarding its effects on cardiorespiratory fitness or HRQoL. No studies to date have compared Nordic Walking with other structured exercise programs. Well-designed, adequately powered, and longer-term RCTs are needed to confirm these preliminary findings and to better define the role of Nordic Walking within evidence-based oncology rehabilitation frameworks.

## Figures and Tables

**Figure 1 cancers-17-03170-f001:**
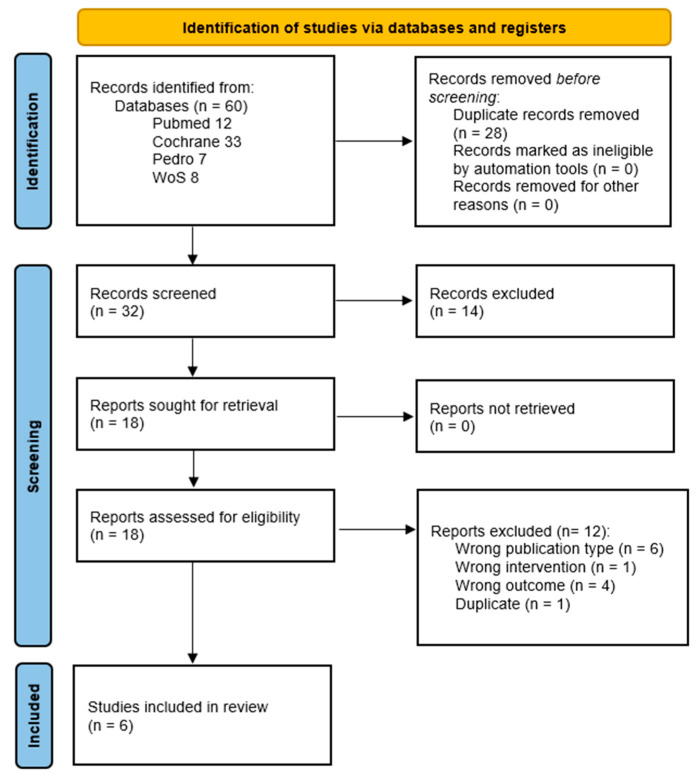
PRISMA flow diagram.

**Figure 2 cancers-17-03170-f002:**
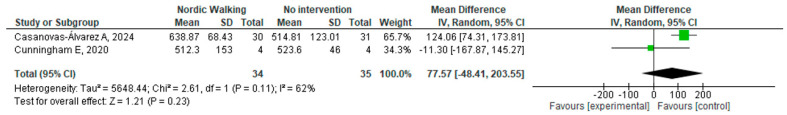
Cardiorespiratory fitness Metanalysis at post-intervention [18,19].

**Figure 3 cancers-17-03170-f003:**
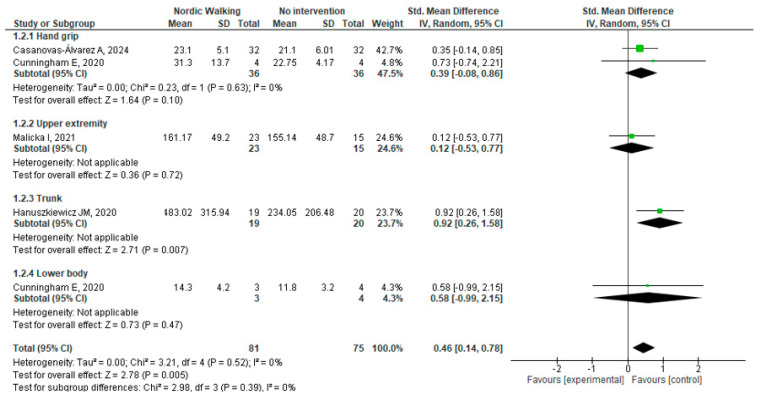
Muscle Strength Metanalysis at post-intervention [18,19,21,22].

**Figure 4 cancers-17-03170-f004:**
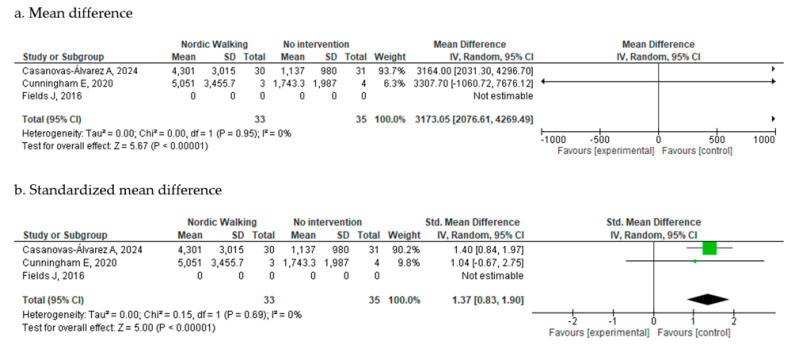
Physical Activity Metanalysis at post-intervention [18,19,20].

## Data Availability

All data is provided in the Appendix A.

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
