# Peer review of "Effects of Nordic Walking on Physical Fitness in Patients with Cancer: A Systematic Review"

_cancers, 2025, doi:10.3390/cancers17193170_

Round 1
Reviewer 1 Report (Previous Reviewer 4)
Comments and Suggestions for Authors
The authors' comments simply haven't convinced me that the addition of walking sticks makes any difference compared to unassisted walking. I just don't see the relevance of this, unless they can cite studies that find that walking sticks prevent falls, or injuries, or other adverse consequence in older adults. Otherwise, I just don't think the study is impactful.
Author Response
Comment: The authors' comments simply haven't convinced me that the addition of walking sticks makes any difference compared to unassisted walking. I just don't see the relevance of this, unless they can cite studies that find that walking sticks prevent falls, or injuries, or other adverse consequence in older adults. Otherwise, I just don't think the study is impactful.
Response:
We thank the reviewer for this valuable comment and acknowledge the concern regarding the relevance of adding poles compared to unassisted walking. In the revised manuscript, we have expanded our rationale for including Nordic Walking (NW) by adding the following points:
-
Physiological and biomechanical rationale: NW differs from regular walking because the active use of poles engages both upper and lower body musculature, thereby increasing energy expenditure [8], improving upper-body strength and posture [9], and enhancing gait parameters such as stride length, gait pattern, and variability [10]. These aspects are particularly relevant for populations with gait instability.
-
Safety and adherence considerations: NW has been reported to improve postural stability [10], which could contribute to safer exercise participation. Moreover, since it is typically practiced outdoors and builds upon walking—a preferred form of exercise among many patients with cancer [12]—NW has the potential to enhance adherence [11].
-
Rehabilitation relevance: Combining accessibility, safety, and comprehensive fitness benefits, NW represents a feasible and attractive modality within oncology rehabilitation and survivorship care.
We have integrated these points into the revised Introduction to clarify the added value of NW compared to unassisted walking and to highlight its potential impact in clinical and rehabilitation contexts.
- Pellegrini, B.; Boccia, G.; Zoppirolli, C.; Rosa, R.; Stella, F.; Bortolan, L.; Rainoldi, A.; Schena, F. Muscular and Metabolic Responses to Different Nordic Walking Techniques, When Style Matters. PLoS One 2018, 13, e0195438, doi:10.1371/journal.pone.0195438.
- Bullo, V.; Gobbo, S.; Vendramin, B.; Duregon, F.; Cugusi, L.; Di Blasio, A.; Bocalini, D.S.; Zaccaria, M.; Bergamin, M.; Ermolao, A. Nordic Walking Can Be Incorporated in the Exercise Prescription to Increase Aerobic Capacity, Strength, and Quality of Life for Elderly: A Systematic Review and Meta-Analysis. Rejuvenation Res 2018, 21, 141–161, doi:10.1089/rej.2017.1921.
- Reuter, I.; Mehnert, S.; Leone, P.; Kaps, M.; Oechsner, M.; Engelhardt, M. Effects of a Flexibility and Relaxation Programme, Walking, and Nordic Walking on Parkinson’s Disease. J Aging Res 2011, 2011, 232473, doi:10.4061/2011/232473.
- Lacharité-Lemieux, M.; Brunelle, J.-P.; Dionne, I.J. Adherence to Exercise and Affective Responses: Comparison between Outdoor and Indoor Training. Menopause 2015, 22, 731–740, doi:10.1097/GME.0000000000000366.
- Wong, J.N.; McAuley, E.; Trinh, L. Physical Activity Programming and Counseling Preferences among Cancer Survivors: A Systematic Review. Int J Behav Nutr Phys Act 2018, 15, 48, doi:10.1186/s12966-018-0680-6.
Reviewer 2 Report (Previous Reviewer 1)
Comments and Suggestions for Authors
To my opinion, the authors revised the manuscript suffciently.
Author Response
We thank the reviewer for their positive feedback and are pleased that the revisions addressed the concerns raised.
Round 2
Reviewer 1 Report (Previous Reviewer 4)
Comments and Suggestions for Authors
Ok, I'm sold. I appreciate the additional references noting the ways in which NW may be better than unassisted walking for some populations of patients. I still think the research is fairly low impact, since most oncology patients are able to walk unassisted, and the bigger challenge is compliance with a walking program. But perhaps this research would be helpful for people with significant instability as a result of their cancer treatment or general deconditioning.
This manuscript is a resubmission of an earlier submission. The following is a list of the peer review reports and author responses from that submission.
Round 1
Reviewer 1 Report
Comments and Suggestions for Authors
- I just have one question about the strength measurements. In Figure 3, the strength measurement data was pooled. What method was used to perform the strength measurements? Was it about maximum strength or strength endurance?
- Another question about the assessment methods. Why did not define certrain assessment methods (for example leg press)?
- How can Nordic walking be used? For which patients is this sport suitable? How can the effort be measured?
Author Response
Comment 1: I just have one question about the strength measurements. In Figure 3, the strength measurement data was pooled. What method was used to perform the strength measurements? Was it about maximum strength or strength endurance?
Response 1: Thank you for your insightful comment. The measurement tools used in each study are detailed in Table 1 and differ across studies. For example, handgrip strength—reported in most studies—was typically assessed using a dynamometer, which measures maximum strength. Other outcomes, such as trunk muscle performance in Hanuszkiewicz et al., were evaluated with endurance-based tests. Due to this heterogeneity in assessment methods, we applied the standardized mean difference in the meta-analysis to allow for pooled comparisons.
Comment 2: Another question about the assessment methods. Why did not define certrain assessment methods (for example leg press)?
Response 2: We intentionally adopted a broad scope for inclusion criteria, as we anticipated a limited number of available studies on this topic. Therefore, we did not restrict the review to specific assessment tools (e.g., leg press), as long as the methods measured strength-related outcomes relevant to our research question. This approach ensured that we could capture as much evidence as possible on the effects of Nordic walking in patients with cancer.
Comment 3: How can Nordic walking be used? For which patients is this sport suitable? How can the effort be measured?
Response 3: We initially planned to conduct subgroup analyses (e.g., according to cancer type, treatment duration, or number of sessions). However, the limited number of eligible studies prevented us from performing these analyses. While the primary aim of our review was not to establish clinical guidelines for Nordic walking, we address its potential applicability, safety considerations, and relevance for different patient groups in the section “Implications for practice, policy, and future research.” In short, Nordic walking appears promising as a feasible and adaptable activity for patients with cancer, but more high-quality studies are needed to clarify for whom and under what conditions it is most effective.
Reviewer 2 Report
Comments and Suggestions for Authors
Comments:
The manuscript entitled “Effects of Nordic Walking on Physical Fitness in Patients With 2 Cancer: A Systematic Review” attempted to emphasize the beneficial effects of Nordic walking on fitness variables. Although synthesized evidence revealed that Nordic walking has some beneficial effects in cancer survivors, there are a lot of issues that should be addressed, and the manuscript needs to be further improved.
Comments:
- The number of included studies is less, and the sample size is small
- The search date, March 2024, is a bit old, more than a year ago. Please update the search and include the most recent studies, if applicable.
- What are the five databases used for searching the articles?
- The total number of studies and participants should be mentioned in the Abstract.
- From the Abstract, it is unclear whether the outcomes results are compared between the trials or within the Nordic walking trial.
- Line 42-43. The authors mentioned that "particularly breast cancer survivors" in the abstract's conclusion. However, corresponding results for this subgroup are not presented in the Results section, which could be misleading or confusing.
- Introduction: In which cancer survivors Nordic walking efficient for improving the physical fitness or strength?
- Is there any prior systematic review and meta-analysis that addresses a similar issue? Please provide a brief background to emphasize the need of conducting this systematic review and meta-analysis.
- Article search details and minimum keywords are missing in the Method section.
- I understand that the authors might have intended to include a larger number of studies through their inclusion criteria. However, it is unclear about the ‘comparator’. Is it acceptable to use both comparators in one analysis?
- Advised to use the Risk of Bias assessment version 2 to assess the quality of the included
- Lines 210-213. Please check and think about the heterogeneity if there are more number of studies with different time points of assessment data.
- Some outcomes are only from one study. How could authors justify such analysis and results?
- According to the PRISMA guideline, what is the minimum number of articles (RCTs) required for meta-analysis?
- Is there any minimum required number of studies for subgroup analysis?
- The search date reported in the Results does not match with the date reported in the Abstract and Methods.
- Figure 2-4 legends are incomplete. Authors should check and write a detailed figure legend before submission.
- Figure 2, It seems the ported cardiorespiratory fitness data should be checked. For the Casanovas-Alvarez et al. study, the control group value of 514.81 does not appear to be reported in their study.
- There is a lot more to improve in the Results and Discussion sections.
Author Response
Commet 1: The number of included studies is less, and the sample size is small
Response 1: We agree with the reviewer. The limited number of studies and small sample sizes are important limitations, which we have explicitly acknowledged in the Limitations section of the Discussion. Unfortunately, this reflects the current state of the evidence, and while we cannot change it, we believe it highlights the need for further high-quality trials in this area.
Comment 2: The search date, March 2024, is a bit old, more than a year ago. Please update the search and include the most recent studies, if applicable.
Response 2: We apologize for the error. The last search update was in November 2024, not March 2024. This has been corrected in the Abstract, Methods, and Results. While several months have passed since then, the time lag between the literature search and publication is common in systematic reviews. During manuscript preparation we also monitored for new publications and are not aware of any relevant studies published since November 2024. A formal update could be performed, but it would substantially delay the submission process.
Comment 3: What are the five databases used for searching the articles?
Respone 3: Thank you for this point. As stated in Section 2.2 (Information sources and search strategies), the databases searched were: PubMed, MEDLINE (via Ovid), Cochrane Library Plus, Web of Science, and PEDro (Physiotherapy Evidence Database).
Comment 4: The total number of studies and participants should be mentioned in the Abstract.
Response 4: The number of included studies is already reported in the Abstract (“This systematic review included six RCTs”). We chose not to add the total number of participants, as these differ by outcome and are described in detail in the Results section. We believe adding this figure in the Abstract may be misleading, but we are open to including it if the Editor considers it necessary.
Comment 5: From the Abstract, it is unclear whether the outcomes results are compared between the trials or within the Nordic walking trial.
Response 5: Thank you for pointing this out. To improve clarity, we have revised the Abstract Objective and Conclusion as follows:
“Objective: To evaluate NW's effects on physical fitness, health-related quality of life (HRQoL), adherence, and safety in patients living with and beyond càncer, compared with no intervention or other exercise programs”
This has also been clarified in the conclusion section of both the abstract and the conclusion section.
“Conclusion: When compared with no intervention, NW is feasible and safe, and may improve muscle strength and physical activity in patients with cancer, particularly breast cancer survivors. Evidence for cardiorespiratory fitness and HRQoL remains inconclusive. To date, no studies have compared Nordic walking with other structured exercise programs. Higher-quality RCTs with diverse populations are needed.”
“Nordic walking appears to be a feasible and low-risk exercise option that may improve muscle strength (low-certainty evidence) and physical activity levels (moderate-certainty evidence) in people with cancer when compared with no intervention. However, current evidence is too limited and heterogeneous to draw firm conclusions regarding its effects on cardiorespiratory fitness or HRQoL. No studies to date have compared Nordic walking with other structured exercise programs. Well-designed, adequately powered, and longer-term RCTs are needed to confirm these preliminary findings and to better define the role of Nordic walking within evidence-based oncology rehabilitation frameworks.”
Comment 6: Line 42-43. The authors mentioned that "particularly breast cancer survivors" in the abstract's conclusion. However, corresponding results for this subgroup are not presented in the Results section, which could be misleading or confusing.
Answer 6: We agree. The reference to breast cancer survivors has been removed from the Abstract conclusion to avoid confusion.
Comment 7: Introduction: In which cancer survivors Nordic walking efficient for improving the physical fitness or strength?
Response 7: Thank you for the question. This is precisely the focus of our systematic review — to synthesize the available evidence across different cancer populations. However, due to the small number of studies, this anaysis was not possible.
Commet 8: Is there any prior systematic review and meta-analysis that addresses a similar issue? Please provide a brief background to emphasize the need of conducting this systematic review and meta-analysis.
Response 8: Yes, this has been addressed in the Introduction and expanded in the Discussion:
“Moreover, only one prior systematic review has synthesized findings related to NW in breast cancer which is now outdated, having been published over five years ago [10]. Given the likely emergence of new evidence across diverse cancer types and fitness-related outcomes, an updated and broader review is warranted”.
Comment 9: Article search details and minimum keywords are missing in the Method section.
Answer 9: Full search strategies for each database are provided in the Supplement (Appendix I).
Comment 10: I understand that the authors might have intended to include a larger number of studies through their inclusion criteria. However, it is unclear about the ‘comparator’. Is it acceptable to use both comparators in one analysis?
Response 10: As explained in the methodology, comparators have been analyzed separately.
“2.7 Data synthesis: Data were analyzed for each comparison outlined in the review (NW vs. no intervention and NW vs. other exercise programs) and for three time points: post-intervention, three months post-intervention, and six months post-intervention. Studies not matching these exact time points were included in the closest category”
Commet 11: Advised to use the Risk of Bias assessment version 2 to assess the quality of the included
Response 11: We acknowledge the usefulness of the RoB 2 tool. However, we opted for the JBI Critical Appraisal Checklist for RCTs, which is also a validated and widely used tool. We have clarified this choice in the Methods.
Comment 12: Lines 210-213. Please check and think about the heterogeneity if there are more number of studies with different time points of assessment data.
Response 12: Thank you for this observation. We have addressed heterogeneity by analyzing outcomes separately for different follow-up times (post-intervention and 6 months). We believe this approach appropriately manages potential variability in time points.
Comment 13: Some outcomes are only from one study. How could authors justify such analysis and results?
Response 13: None of the meta-analyses included only one study. In some subgroup analyses, individual studies are presented separately (e.g., by muscle group) to explore possible differences across outcomes. These are presented descriptively rather than pooled when only one study was available.
Comment 14: According to the PRISMA guideline, what is the minimum number of articles (RCTs) required for meta-analysis?
Response 14: PRISMA is a reporting guideline and does not specify a minimum number of studies for meta-analysis. It aims to ensure transparency and completeness of reporting rather than methodological thresholds.
Comment 15: Is there any minimum required number of studies for subgroup analysis?
Response 15: There is no strict minimum required. Subgroup analyses are exploratory and help to identify potential sources of heterogeneity, even when based on a limited number of studies.
Comment 16: The search date reported in the Results does not match with the date reported in the Abstract and Methods.
Response 16: We apologize for the inconsistency. The correct date is November 2024. This has been corrected in all sections.
Comment 17: Figure 2-4 legends are incomplete. Authors should check and write a detailed figure legend before submission.
Response 17: Thank you for the comment. We are unclear about which elements of the Figure 2–4 legends you consider incomplete. Could you please specify the missing information so we can revise the legends accordingly?
Comment 18: Figure 2, It seems the ported cardiorespiratory fitness data should be checked. For the Casanovas-Alvarez et al. study, the control group value of 514.81 does not appear to be reported in their study.
Response 18: We confirm this value. As described in the Methods, we contacted study authors when data were missing. Casanovas-Alvarez et al. provided us with the control group value directly. We have clarified this in the manuscript.
Comment 19: There is a lot more to improve in the Results and Discussion sections.
Response 19: Thank you for this observation. We would appreciate more specific guidance or examples regarding which aspects of the Results and Discussion sections require improvement, so that we can address them more effectively.
Reviewer 3 Report
Comments and Suggestions for Authors
Introduction: The introduction is comprehensive and well-referenced. It presents the rationale, context, and relevance of the study. The justification for updating previous reviews is satisfactory, especially considering the emerging evidence in oncology regarding physical exercise.
Methods: Adherence to the PRISMA and GRADE guidelines is a positive aspect. However, some methodological aspects could be more transparent. Please provide the full search dates and search strategy strings directly in the main manuscript (and not only in the supplementary material).
The PICO criteria are well described, although the definition of quasi-randomized trials and how they were handled (especially in quality assessment) could be better clarified.
The decision to include composite scores (e.g., for muscle strength) through the aggregation of means across different movements should be justified from an analytical or methodological standpoint.
More details are needed regarding the handling of missing data (e.g., was any type of imputation used?).
Consider describing whether publication bias was assessed using methods beyond funnel plots (e.g., Egger’s test), even if only narratively, since the number of included studies is fewer than 10.
Results: The results are well organized, with appropriate subgroup and sensitivity analyses. The meta-analytic approach is well described; however, some numerical inconsistencies could be clarified:
The simultaneous use of mean difference and standardized mean difference in the analysis of physical activity may confuse readers. Consider aligning the narrative to avoid duplication.
Tables 1 and 2 could be merged or better cross-referenced, as they conceptually overlap. But I understand the limitation in presenting them on a single page.
The findings regarding health-related quality of life (HRQoL) are inconclusive due to the heterogeneity of the instruments used and insufficient reported data. This limitation should be emphasized earlier in the Results section, not only in the Discussion.
Discussion: The interpretation of non-significant results for cardiorespiratory endurance is appropriate and cautious. The authors assertively highlight the clinical relevance despite the lack of statistical significance.
However, the extrapolation of findings to broader oncological populations should be done more cautiously. Although Nordic Walking (NW) appears promising, current evidence is concentrated in women with breast cancer.
The authors could better differentiate between the concepts of effectiveness and efficacy, especially when discussing adherence and long-term implications.
Comments on the Quality of English Language
Overall, the manuscript is well written. However, some minor issues persist: The term “muscle streangth” in Figure 3 contains a typographical error and should be corrected to “muscle strength”; There are several instances of redundant phrases (e.g., “feasible and low-risk” appears frequently). A careful editorial review is recommended to improve the fluency and conciseness of the text.
Author Response
Comment 1: Introduction: The introduction is comprehensive and well-referenced. It presents the rationale, context, and relevance of the study. The justification for updating previous reviews is satisfactory, especially considering the emerging evidence in oncology regarding physical exercise.
Response 1: Thank you for this positive feedback.
Comment 2: Methods: Adherence to the PRISMA and GRADE guidelines is a positive aspect. However, some methodological aspects could be more transparent. Please provide the full search dates and search strategy strings directly in the main manuscript (and not only in the supplementary material).
Response 2: Thank you for the suggestion. The full search strings are currently provided in the Supplement, which is freely accessible and allows for detailed reporting without affecting the readability of the main manuscript. However, we are open to moving this information into the main text if the Editor considers it preferable.
Comment 3: The PICO criteria are well described, although the definition of quasi-randomized trials and how they were handled (especially in quality assessment) could be better clarified.
Response 3: Thank you for this comment. By quasi-randomized trials, we refer to studies that described allocation procedures that were not truly random (e.g., by date of birth or order of admission). These were assessed using the same JBI checklist, with the limitations of their allocation method taken into account. We can expand this clarification in the Methods if considered necessary.
Comment 4: The decision to include composite scores (e.g., for muscle strength) through the aggregation of means across different movements should be justified from an analytical or methodological standpoint.
Response 4: Thank you for raising this point. We created composite scores in some cases to provide a more comprehensive estimate of overall muscle strength when individual studies reported multiple strength outcomes (e.g., upper and lower body). This approach reduces the risk of overweighting studies that reported more measures than others. We will add a justification for this methodological decision in the Methods section to make it clearer for readers.
Comment 5: More details are needed regarding the handling of missing data (e.g., was any type of imputation used?).
Response 5: We appreciate this comment. As described in the Methods, in case of missing data we attempted to contact the original study authors. If no response was received, the study was excluded from the corresponding analysis. No imputation methods were applied. “In case of missing data, we tried to contact the authors, if no answer was given, that particular study was not included in the analysis.”
Comment 6: Consider describing whether publication bias was assessed using methods beyond funnel plots (e.g., Egger’s test), even if only narratively, since the number of included studies is fewer than 10.
Response 6: Thank you. As described in the Methods, we planned to use only visual inspection of funnel plot asymmetry when ≥10 studies were available.
Comment 7; Results: The results are well organized, with appropriate subgroup and sensitivity analyses. The meta-analytic approach is well described; however, some numerical inconsistencies could be clarified:
The simultaneous use of mean difference and standardized mean difference in the analysis of physical activity may confuse readers. Consider aligning the narrative to avoid duplication.
Response 7 Thank you for pointing this out. In line with our protocol, mean differences were the primary analysis. However, due to the wide variability in measurement scales, the forest plot was difficult to interpret. For this reason, we additionally calculated standardized mean differences to improve visualization and interpretability, as described in the Results: “Due to the wide range of values observed, an additional analysis using Std.MD was performed for improved visualization and interpretability (Figure 4b), which yielded consistent results”
Comment 8: Tables 1 and 2 could be merged or better cross-referenced, as they conceptually overlap. But I understand the limitation in presenting them on a single page.
Response 8: Thank you for the suggestion. We considered merging them, but due to space constraints and to improve readability we decided to keep them separate. We believe the current format presents the information more clearly.
Comment 9: The findings regarding health-related quality of life (HRQoL) are inconclusive due to the heterogeneity of the instruments used and insufficient reported data. This limitation should be emphasized earlier in the Results section, not only in the Discussion.
Comment 9: Thank you. We agree and note that this is already mentioned in the Results: “HRQoL was assessed in four studies post-intervention [15–17,20]. However, meta-analysis could not be performed due to variability in reporting and lack of sufficient quantitative data. Specifically, Cunningham et al. [16], Fields et al. [17], and Rösner et al. [20] utilized the SF-36 questionnaire but did not report the Physical Component Summary (PCS) scores nor sufficient domain-level data to allow for calculation of a composite score.”
Comment 10: Discussion: The interpretation of non-significant results for cardiorespiratory endurance is appropriate and cautious. The authors assertively highlight the clinical relevance despite the lack of statistical significance.
However, the extrapolation of findings to broader oncological populations should be done more cautiously. Although Nordic Walking (NW) appears promising, current evidence is concentrated in women with breast cancer.
Response 10: We agree with this comment. To address it, we have revised the Conclusion to more cautiously reflect the populations studied:
“Nordic walking appears to be a feasible and low-risk exercise option that may improve muscle strength (low-certainty evidence) and physical activity levels (moderate-certainty evidence) in people with cancer when compared with no intervention. However, current evidence is too limited, heterogeneous, and largely concentrated in breast cancer populations to draw firm conclusions regarding its effects on cardiorespiratory fitness or HRQoL. No studies to date have compared Nordic walking with other structured exercise programs. Well-designed, adequately powered, and longer-term RCTs are needed to confirm these preliminary findings and to better define the role of Nordic walking within evidence-based oncology rehabilitation frameworks.”
Comment 11: The authors could better differentiate between the concepts of effectiveness and efficacy, especially when discussing adherence and long-term implications.
Answer 11: Thank you for raising this point. We would appreciate clarification on whether the reviewer is suggesting that we more explicitly distinguish between: (a) efficacy — the effect of Nordic walking under controlled trial conditions, and (b) effectiveness — its impact in real-world practice, including adherence and sustainability. If so, we can incorporate this distinction into the Discussion.
Reviewer 4 Report
Comments and Suggestions for Authors
This study is valid, and the methods are well planned. However, it adds very little to the existing literature on the impact of exercise on cancer outcomes. The addition of walking sticks (nordic walking-NW) doesn't add much to data investigating the benefits of exercise on cancer-related outcomes. It has been very well proven that almost any type of exercise will benefit strength and endurance and decrease fatigue in cancer survivors. The difficulty is making such exercise programs operationalized amongst cancer survivors. More attention should be paid to intervention studies attempting to make exercise interventions well tolerated, and standard of care for all cancer patients.
Author Response
Comment 1: This study is valid, and the methods are well planned. However, it adds very little to the existing literature on the impact of exercise on cancer outcomes. The addition of walking sticks (nordic walking-NW) doesn't add much to data investigating the benefits of exercise on cancer-related outcomes. It has been very well proven that almost any type of exercise will benefit strength and endurance and decrease fatigue in cancer survivors. The difficulty is making such exercise programs operationalized amongst cancer survivors. More attention should be paid to intervention studies attempting to make exercise interventions well tolerated, and standard of care for all cancer patients.
Response 1: We thank the reviewer for this thoughtful comment and agree that the general benefits of exercise in oncology are already well established. However, our aim was not to re-demonstrate the overall effects of exercise, but rather to systematically examine the specific role of Nordic walking. NW may provide additional value because it is a low-cost, accessible, outdoor-based intervention that combines aerobic and strength elements, and can be easily integrated into daily life. These features may enhance feasibility, adherence, and long-term implementation compared with other exercise modalities.
We fully agree with the reviewer that operationalization and tolerability are central challenges in oncology exercise. In fact, one of the potential advantages of NW is its acceptability among older adults and those with limited exercise experience, which may support its translation into routine care.